Effects of fertilizer and biochar applications on the relationship among soil moisture, temperature, and N2O emissions in farmland

Wang Xiao 1
Lu Ping 2
Yang Peiling cau_yangpeiling@163.com 1
Ren Shumei 1
1 College of Water Resources and Civil Engineering, China Agricultural University , Beijing , China
2 College of Sericulture, Textile and Biomass Sciences, Southwest University , Chongqing , China
Ahmed Mukhtar
Electronic publication date: 2021 Jul 20
Publication date: 2021
Volume: 9
Electronic Location ID: e11674
Received 2020 Dec 16; Accepted 2021 Jun 4
Copyright: ©2021 Wang et al.
Copyright year: 2021
Copyright holder: Wang et al.
License: This is an open access article distributed under the terms of the Creative Commons Attribution License, which permits unrestricted use, distribution, reproduction and adaptation in any medium and for any purpose provided that it is properly attributed. For attribution, the original author(s), title, publication source (PeerJ) and either DOI or URL of the article must be cited.
License URL: https://creativecommons.org/licenses/by/4.0/

Keywords: Biochar, N2O emissions, Soil moisture, Soil temperature, Fertilization, Sensitivity coefficient, Exponential fitting, Multivariate nonlinear fitting

Funding: The National Key Research and Development Plan 2019YFC0408700 This work was supported by the National Key Research and Development Plan (2019YFC0408700). The funders had no role in study design, data collection and analysis, decision to publish, or preparation of the manuscript.

==============================
Background

Di-nitrogen oxide (N2O) emissions from soil may lead to nonpoint-source pollution in farmland. Improving the C and N content in the soil is an excellent strategy to reduce N2O emission and mitigate soil N loss. However, this method lacks a unified mathematical index or standard to evaluate its effect.

Methods

To quantify the impact of soil improvement (C and N) on N2O emissions, we conducted a 2-year field experiment using biochar as carbon source and fertilizer as nitrogen source, setting three treatments (fertilization (300 kg N ha−1), fertilization + biochar (30 t ha−1), control).

Results

Results indicate that after biochar application, the average soil water content above 20 cm increased by ∼26% and 26.92% in 2019, and ∼10% and 12.49% in 2020. The average soil temperature above 20 cm also increased by ∼2% and 3.41% in 2019. Fertigation significantly promotes the soil N2O emissions, and biochar application indeed inhibited the cumulation by approximately 52.4% in 2019 and 33.9% in 2020, respectively. N2O emissions strongly depend on the deep soil moisture and temperature (20–80 cm), in addition to the surface soil moisture and temperature (0–20 cm). Therefore, we established an exponential model between the soil moisture and N2O emissions based on theoretical analysis. We find that the N2O emissions exponentially increase with increasing soil moisture regardless of fertilization or biochar application. Furthermore, the coefficient a < 0 means that N2O emissions initially increase and then decrease. The aRU < aCK indicates that fertilization does promote the rate of N2O emissions, and the aBRU > aRU indicates that biochar application mitigates this rate induced by fertilization. This conclusion can be verified by the sensitivity coefficient (SCB of 1.02 and 14.74; SCU of 19.18 and 20.83). Thus, we believe the model can quantify the impact of soil C and N changes on N2O emissions. We can conclude that biochar does significantly reduce N2O emissions from farmland.

Introduction

Soil N2O emissions, representing a significant N loss, are inevitable products of chemical fertilizer application (Zou et al., 2005). Based on statistics, greenhouse gas emissions from agricultural sources account for 11% of the global greenhouse gas emissions, which exceed the 2020 emission target (Zhao et al., 2016). It was considered that the improvement of soil C and N content (straw returning, biochar application, etc.) is an excellent strategy to reduce N2O emission and mitigate soil N loss. Straw returning significantly mitigated annual N2O emission (Yao et al., 2017; Zhou et al., 2017); and compared with the nitrogen fertilizer treatment, the reduction of N2O emissions due to straw returning treatment could be as much as 35% in a particular year (Hu et al., 2016). Biochar application could suppress N2O production by 91% in near-saturated soils (Case et al., 2015). With the considerable amount of biochar application, the N2O emissions decreased sharply (Bruun et al., 2011b). Slow-release and controlled-release fertilizer can also help lower N2O emissions in farmland (Bordoloi & Baruah, 2016; Braun & Bremer, 2018; Vico et al., 2020). The slow-release fertilizer treatments significantly decreased N2O emissions by 16.94–21.20% for the rice-wheat cropping system in eastern China (Shakoor et al., 2018). The amendment with a controlled-release fertilizer in soil could help reduce N2O emissions by 26–50% than urea application without sacrificing grain yield (Ji et al., 2012). There are many studies on soil improvement using C and N (Ali et al., 2021; Cheng, 2020; Liu et al., 2021; Pal & Marschner, 2016); however, they lack a unified mathematical index or standard to evaluate the effect of the research done. The differences of objective conditions in each study make it difficult to draw horizontal comparison of results, which leads to the inability to judge the real effect of soil improvement. A mathematical model or index is required according to the field measured data, to form a unified standard to judge the merits of soil C and N improvement, which will be very conducive to the promotion of soil C and N improvement in farmland and the sustainable development of agriculture.

Therefore, some more accurate methods, such as mathematical models, are still needed to quantify the impact of soil C and N improvement on N2O emissions. In addition to increasing soil C and N, straw returning also increased other nutrient elements (Li, Zhao & Huang, 2002). We made an improvement in soil C and N using biochar and chemical fertilizer application to achieve the controlling variables. Biochar, a form of exogenous carbon, is produced by pyrolysis of straw and branches. The amendment with biochar remarkably affects the physical soil properties (Nanda et al., 2016) and biochemical reactions (Gul et al., 2015; Henrique et al., 2015) and thus affects C and N cycles in soil (Liu et al., 2021).

Biochar amendment of soil has many physical effects; for example, it improves the water holding capacity of the soil (Major et al., 2012), nitrate retention (Ghulam et al., 2017; Zhang et al., 2010), and soil aeration (Alfred et al., 2018). The improvement of the soil’s water holding capacity due to biochar application is the primary factor inhibiting N2O emissions (Basso et al., 2013). Soil moisture affects the production of N2O and the conversion of N2O to N2. The N2O emissions increases and reach a plateau when the water-filled-pore-space (WFPS) is ∼60%–70%. At the same time, denitrification was maximum. When WFPS was near-saturation, a more anaerobic environment, the N2O emissions decrease (Prado et al., 2006). The heat absorption ability of biochar can improve the soil ambient temperature and soil microbial activity. The soil temperature, another critical factor, directly influences the activities of the nitrifying and denitrifying microorganisms and urea hydrolysis (Alvarez et al., 2018). With the increase of soil temperature, microorganisms became more active, biochemical reaction speed increased rapidly, and the efficiency of N2O production under the same reaction substrate increased significantly (Case et al., 2012). Compared with the indirect influence of biochar on N2O emissions, the fertilizer application added N2O reaction substrate, which significantly increased the production efficiency of N2O. With the substrate increase, the N2O sensitivity with soil moisture and temperature increased significantly, changing the N2O emission pattern. Many studies set up multiple groups of C and N treatments in the laboratory to establish the relationship between soil C and N content and N2O emission (Feng & Zhu, 2017; Horák et al., 2017; Zwieten et al., 2014).

However, we have made it clear that biochar does not directly affect N2O emissions, and it lacks physical significance to establish a function between biochar concentration and N2O emissions. Additionally, the N2O emission observation made through laboratory tests cannot accurately reflect the actual situation of farmland N2O emission. The model just including soil C content (or N content) and N2O emission is not widely applicable without considering the effect of soil moisture (or soil temperature).

Therefore, to establish an appropriate standard for evaluating the benefits of soil C and N improvement, we conducted a 2-year field experiment using biochar as a carbon source and fertilizer as a nitrogen source. The central hypothesis was that the trend of N2O emission under different soil C and N levels predicted by measuring soil moisture and temperature could be found out. We hope to (1) explore the response of soil N2O emissions to soil moisture and temperature under different conditions (no fertilization, fertilization, fertilization + biochar, and (2) build models and quantify the effects of fertilization and biochar application on N2O emissions.

Materials & Methods

Experimental site

The experiments were conducted between 2019–2020 at the experimental station of the China Agricultural University, China (latitude: 39°42′07.8″N, longitude: 116°41′48.0″E, altitude: 24 m) in loam soil (9.6% clay, 52.6% silt, and 37.8% sand). The mean temperature and precipitation were 26.6 °C and 358 mm, respectively, in 2019 (April to September); The mean temperature and precipitation were 28.4 °C and 377 mm, respectively, in 2020 (July to September). The experimental soil had a soil bulk density of 1.38 g cm−3, and the field capacity of the 0–20 cm soil layer was 22.87%, according to the method from Grossman & Reinsch (2002). The soil (0–20 cm) had an NH4+-N of 6.31 mg kg−1, NO3−-N of 29.00 mg kg−1.

Experimental design

This experiment aims to establish the relationship between farmland soil moisture, temperature, and N2O emissions and explore whether changes in soil C and N content can affect N2O emissions. Therefore, long-term monitoring of soil moisture, temperature, and N2O emissions under different soil C and N conditions are required. To make the effect more realistic, we chose maize crop to carry out the research. The maize cultivar Zhengdan 958 is widely used in China. Maize was planted on April 10, 2019, and June 15, 2020, with a 0.5 m line spacing and 0.3 m between plants and harvested on September 5 in 2019 and September 25 in 2020. There is a 1-m-wide transition zone between adjacent plots. To creat varying soil C and N contents, we set three treatments in the experiment: (1) RU: fertilization with urea, irrigation with reclaimed water; (2) BRU: soil amendment with 30 t ha−1 biochar, fertilization with urea, and irrigation with reclaimed water; and (3) CK: irrigation with reclaimed water. During the maize growth period, soil moisture and temperature were monitored daily at a depth of 80 cm (0–10 cm, 10–20 cm, 20–40 cm, 40–60 cm, 60–80 cm) and N2O emission flux in the soil surface was detected every 2-3 days. Then, we explore the correlation between the three variables so as to establish a mathematical model.

The N fertilizer (urea; 300 kg N ha−1) was applied as follows: 40% before sowing, 30% during the silking stage, and 30% during the filling stage. The P (calcium superphosphate; 40 kg P ha−1) and K (potassium sulfate; 80 kg K ha−1) fertilizers were applied before sowing. The total irrigation amount for each treatment was 230 mm in 2019 and 250 mm in 2020, respectively.

The soil temperature and water content (0–80 cm) were measured with an ET-100 (Insentek, China). The WFPS was calculated with the following equation:

(1) WFPS=θm⋅ρ0ρH2O⋅ρs⋅100%,

where θm is the gravimetric water content (mg mg−1), ρ0 is the bulk soil density (mg m−3), ρH2O is the density of water (mg m−3), and ρ s is soil particle density (mg m−3).

Biochar was produced by pyrolysis (450 °C) of maize straw and used for the field experiment. The biochar had a pH of 8.2, total C content of 657 g kg−1, total N content of 9 g kg−1, available K of 16 g kg−1, available P of 0.8 g kg−1, and density of 0.297 g cm−3. The biochar was evenly applied to the surface soil (30 t ha−1; top 20 cm of the soil) in April 2019 before sowing maize. The initial soil had a pH of 9.86, soil organic carbon (SOC) of 29.71 g kg−1, soil organic nitrogen (SON) of 2.3, NH4+-N of 6.48 mg kg−1, NO3−-N of 28.00 mg kg−1, available K of 38.14 g kg−1, and available P of 1.70 g kg−1.

Gas collection and analysis

The N2O fluxes were measured at every plot using a static closed chamber method (Qi et al., 2015). The sampling chamber consisted of two parts: a soil ring without top and bottom (50 cm in diameter and 30 cm high) and a removable cover (50 cm in diameter and 50 cm high). The soil ring was directly inserted into the soil approximately 25 cm below the soil surface, leaving five cm from the soil surface. The removable cover was placed on top during the sampling and removed afterward. Two fans with diameters of 10 cm were installed on the sidewall of each cover to create turbulent airflow when the chamber was closed. Three gas samples were obtained during each treatment and sampling period,i.e., three replicates of one treatment. The soil temperature from 9:00 to 11:00 A.M. was close to the daily mean soil temperature. Thus, we took gas samples during this period. The air temperature inside the static closed chamber was also measured. Gas samples (50 ml each) were collected in four times intervals (0, 10, 20, and 30 min) using 50 ml plastic syringes. The N2O fluxes were measured after rainfall, fertilization, or every two days. The N2O was analyzed using a gas chromatograph (GC 7890 A; Agilent, Santa Clara, CA, USA) and electron capture detector (ECD) within 48 h. The N2O daily emissions were calculated with the following equation:

(2) F=ρ×V∕A×dc∕dt×273∕273+T,

where F is the N2O flux (g m−2 h−1), ρ is the density of the gas in a standardized state (g m−3), V is the volume of the chamber (m3), A is the cross-sectional area of the chamber (m2), dc/dt is the rate of gas accumulation (µg kg−1 h−1), and T is the chamber temperature (∘C).

The cumulative N2O emissions (kg ha−1) were calculated by using the linear interpolation method.

Chemical analyses

The pH values of the biochar and soil were determined with a pH electrode (420A plus; Thermo Orion). The biochar/deionized water and soil/deionized water ratios were 1:30 w/w and 1:10 w/w, respectively, after being stirred for 1.5 min and equilibrated for 1 h. The C and N concentrations of the biochar and soil were determined using an elemental analyzer (Flash 2000; Thermo Fisher, Waltham, MA, USA). The available P content was measured with an ultraviolet–visible spectrophotometer (TU-1901 UV–Vis; Beijing Puxi Instrument Company, Beijing, China). The available K content was measured with a flame photometer (FAAS; Zenit 700P; Analytik Jena AG, Jena, Germany). The NH4+-N and NO3−-N concentrations were measured using segmented flow analysis (SFA; Futura, Alliance, France).

Model

N2O emissions are the result of soil biochemical reactions, which are slow. N2O emission at a particular time may arise from the cumulative effects of water and temperature in the previous period. Therefore, the lag effects should be considered. We assume that the N2O emission during sampling is caused by the influence of soil moisture and temperature in the previous 24 h. If sampling occurs at 9:00 am on July 23, the N2O flux is affected by the soil moisture (or temperature) between 10:00 A.M. on July 22, and 9:00 A.M. on July 23. ET-100 can monitor a series of soil moistures and temperatures hourly. Therefore, we established a function between N2O flux and the average soil moisture (or temperature) in the past 24 h. This function was created to calculate the daily soil moisture and temperature at 9:00 as the node.

A) Relationship among the soil temperature, water content, and N2O emissions

The principal component analysis is a statistical method. It transformed a group of correlated variables into a group of linearly unrelated variables based on orthogonal transformation. The transformed variables are called principal components (Stacklies et al., 2007). Through principal component analysis, we have synthesized numerous indexes and eliminated the sample’s overlapping (Granato et al., 2018; He, Mao & Han, 2018; Imaizumi & Kato, 2018). The expression of the principal component was as follows: (3) W1=a⋅W10∗+b⋅W20∗+c⋅W40∗

(4) T1=d⋅T10∗+e⋅T20∗+f⋅T40∗,

Table 1 Correlation among the soil water content, temperature, and N2O emissions.

		Soil water content in different depth	
		10 cm	20 cm	40 cm	60 cm	80 cm	
	RU	0.564**	0.761**	0.465**	0.097	−0.310**	
2019	BRU	0.767**	0.883**	0.704**	0.427**	0.008	
	CK	0.834**	0.906**	0.701**	0.557**	0.341**	
	RU	0.886**	0.881**	0.423**	0.011	−0.120**	
2020	BRU	0.783**	0.805**	0.641**	−0.514**	−0.747**	
	CK	0.389**	0.775**	0.300**	−0.092	−0.083	
		Soil temperature in different depth	
		10 cm	20 cm	40 cm	60 cm	80 cm	
	RU	0.377**	0.437**	0.502**	0.494**	0.438**	
2019	BRU	0.667**	0.751**	0.309**	0.451**	0.478**	
	CK	0.087	0.274*	0.529**	0.666**	0.670**	
	RU	0.496**	0.551**	0.501**	0.494**	0.380**	
2020	BRU	0.568**	0.546**	0.512**	0.403**	−0.075	
	CK	0.297**	0.215*	0.327**	0.100	−0.076	
Notes.

*, ** Significant at P < 0.05, 0.01 levels, respectively (least significant difference test).

where W1 or T1 is the principal components obtained by extracting the soil water content or temperature from the 10, 20, and 40 cm soil layers, respectively; W10 ∗, W20∗, W40 ∗, T10∗, T20 ∗, and T40∗ are the standardized soil water contents and temperatures corresponding to the 10, 20. and 40 cm soil layers, respectively; and a, b, c, d, e, and f are the standardized coefficients of the values, respectively.

Because the principal component 1 (PC1) accounts for more than 70% of the variation, only PC1 of the soil moisture content (or temperature) was used for multivariate nonlinear fitting. Table 1 shows that the PC1 covers the soil moisture content and temperature information of the 10, 20, and 40 cm soil layers. The moisture content and temperature of the 20 cm soil layer contribute the most to the PC1.

The parameter F∗N was obtained by standardizing the daily N2O emissions and fitting with W1 and T1: (5) FN∗W1,T1=z0+k1∗W1+k2∗T1+k3∗W12+k4∗T12+k5∗W1⋅T1,

where k1 (k2, k3, k4, k5) is the coefficient and z0 is the constant.

Table 2 shows the value of the coefficient in Eq. (5).

Table 2 Coefficients of PC1.

	a	b	c	d	e	f	
2019	0.610	0.537	0.583	0.596	0.607	0.526	
2020	0.622	0.615	0.486	0.585	0.573	0.573	
Notes.

The coefficients above represents using MNF-DR analyzed the total observed points (RU+BRU+CK).

Equations (3) and (4) were substituted into Eq. (5) to obtain: (6) F∗W10∗,W20∗,W40∗,T10∗,T20∗,T40∗=z0+k1⋅a⋅W10∗+k1⋅b⋅W20∗+k1⋅c⋅W40∗+k2⋅d⋅T10∗+k2⋅e⋅T20∗+k2⋅f⋅T40∗+k3⋅a⋅W10∗+b⋅W20∗+c⋅W40∗2+k4⋅d⋅T10∗+e⋅T20∗+f⋅T40∗2+k5⋅a⋅W10∗+b⋅W20∗+c⋅W40∗⋅d⋅T10∗+e⋅T20∗+f⋅T40∗

B) Response of the N2O emissions to WFPS

We assumed that the N2O emissions exponentially increase with increasing WFPS and that the emission rate of N2O initially increases and then decreases. Thus, dDE/ dW initially is positive and then negative. The model of the N2O emissions can be obtained as follows: (7) 1DEdDEdW=B−AW,

where DE represents the daily emissions of N2O (kg hm−2), W is the WFPS, and A and B are constants.

To illuminate the mitigation of the N2O emissions due to biochar amendment, we adopted the sensitivity coefficient (SC) to express the effect of the change in the soil water content on the N2O emissions (Tan, Cui & Luo, 2017). The smaller SC is, the smaller is the response of the N2O emissions to the change in the soil water content. The SC can be calculated as: (8) SC=∑ΔDE∕DE0 ∑ΔW∕W0,

where ΔDE is the variation in the N2O emissions between the BRU/RU and CK treatments, ΔW is the variation in the WFPS between the BRU/RU and CK treatments, and DE0 and W0 represent the N2O emissions and WFPS of treatment CK, respectively.

Equation (9), obtained by integrating Eq. (7), is a numerical model describing the increase in the N2O emissions for different WFPS values under irrigation. (9) DE=eaW2+bW+c,

where a is A/2, b is B, and c is an integral constant.

Statistical analysis

The data were analyzed with SPSS20.0 software. Variance analysis (ANOVA) was carried out by using the General Linear Model Univariate procedure. The analysis of significant differences (p < 0.05) between treatments was carried out using Tukey’s range test. We also prepared figures and fitted the models using OriginPro 2019.

Results

Soil water content and temperature

The experimental area was irrigated with 60 mm reclaimed water after sowing. The next irrigation step was conducted at the seeding stage. We observed a drastic fluctuation in the soil water content above a depth of 20 cm during each treatment. Such a fluctuation did not occur in the 60-cm and 80-cm soil layers (Fig. 1). In 2019, the average soil water content above 20 cm in the BRU treatment was ∼26% and 26.92% higher than that of the RU and CK treatments, respectively. And The average soil water content (0–20 cm) in BRU also showed this trend in 2020: it was ∼10% and 12.49% higher than those in RU and CK treatments, respectively. However, the difference in soil water content between these treatments was not pronounced below a depth of 40 cm. Many researchers have reported that biochar, owing to the great voids on the surface, can improve soil moisture content (Taghizadeh-Toosi et al., 2011; Qu et al., 2020; Zhang et al., 2019b). Thus, we can conclude that the soil amendment with biochar significantly promotes the water holding capacity.

Figure 1 The soil water content for each treatment in the depth of 10 cm, 20 cm, 40 cm, 60 cm, and 80 cm during 2019–2020 is presented.

(A), (B), and (C) show the soil water content in RU, BRU, and CK treatments in 2019, respectively; (D), (E), and (F) show the soil water content in RU, BRU, and CK treatments in 2020 respectively.

The variation in soil temperature in the maize growth stage for each treatment is shown in Fig. 2. Obviously, the temperature of the surface soil fluctuates more dramatically than that of the deep soil. The accumulative temperature difference between BRU and RU at a depth of 40 cm were −4 °C and −11 °C in 2019 and 2020, respectively (Fig. 2). The average soil temperature for BRU treatment was obviously lower than that for RU treatment below 40 cm in 2020 (Fig. 2). These results imply that biochar inhibits temperature transfer from the surface to deep soil. In 2019, the average soil temperature above 20 cm of the BRU treatment was ∼2% and 3.41% higher than that of the RU and CK treatments, respectively. However, in 2020, the BRU treatment showed no warming effect and reduced water holding capacith. It may be caused by biochar aging (Zhang et al., 2013).

Figure 2 The soil temperature for each treatment in the depth of 10 cm, 20 cm, 40 cm, 60 cm, and 80 cm during 2019–2020 is presented.

(A), (B), and (C) show the soil temperature for RU, BRU, and CK treatment in 2019, respectively; (D), (E), and (F) show the soil temperature for RU, BRU, and CK treatment in 2020.

Soil N2O emissions

Chemical fertilizer application significantly promotes the soil N2O emissions (Fig. 3). The significant difference in N2O emissions between these treatments occurred between June 13 and July 3 in 2019 and on July 29 and August 28 in 2020. The emission flux of N2O was significantly higher for RU than for BRU and CK at this time. The cumulative N2O emissions for the RU treatment are 3.61 kg ha−1 compared with 1.72 and 1.59 kg ha−1 for the BRU and CK treatments, respectively, in 2019. The increment of N2O emissions for RU treatment was 4.54 kg ha−1 compared with 3.00 and 0.78 kg ha−1 for the BRU and CK treatments, respectively, in 2020. Thus, we could find that fertilization significantly enhanced N2O emissions for the two years and biochar application alleviated this trend. Our results are consistent with those of many previous studies, showing that biochar can indeed inhibit N2O emissions from farmland (Bruun et al., 2011a; Cayuela et al., 2014; Takakai et al., 2019).

Figure 3 N2O emissions in the maize growth stage is presented.

(A) and (B) show the N2O emissions in 2019 and 2020, respectively.

The correlation among the soil water content, temperature, and N2O emissions

The soil N2O emissions were strongly correlated with the soil water content at a depth above 40 cm in all the treatments (Table 1). The correlation between the soil N2O emissions and soil temperature was also pronounced at depth above 40 cm for RU and BRU. Both soil water content and temperature affect the soil N2O emissions. Thus, it is imperative to analyze the coupled effect of the soil water content and temperature on the N2O emissions.

Establishing function among soil water content, temperature, and N2O emissions

Table 1 shows that soil water content and temperature at a depth above 80 cm affect the N2O emissions. Soil water content, temperature and N2O emissions are strongly correlated at depths of 0–40 cm, while the correlation is weak in the 60–80 cm soil layer. Therefore, we set up a function model for soil water content, temperature and N2O emissions above 40 cm. To simplify the calculation, we performed principal components analysis on moisture content and temperature in the 10, 20, and 40 cm soil layers. Table 2 and Table 3 show the value of coefficient in Eqs. (3)–(5), respectively.

Table 3 Coefficients of multiple nonlinear regression.

	Treatment	z0	k1	k2	k3	k4	k5	R2	F	
	Total	0.023	0.276	0.301	−0.030	0.014	0.049	0.51	39.37*	
2019	RU	−0.046	0.433	0.196	0.039	−0.022	0.058	0.61	19.91*	
	BRU	−0.309	0.480	0.203	0.093	0.061	−0.044	0.73	33.34*	
	CK	−0.186	0.529	0.146	0.054	0.017	0.015	0.84	65.22*	
	Total	0.008	0.482	0.109	−0.016	−0.002	0.039	0.60	64.26*	
2020	RU	0.022	0.492	0.056	0.065	−0.031	−0.054	0.80	55.30*	
	BRU	0.098	0.490	0.208	−0.003	−0.015	0.045	0.75	43.45*	
	CK	0.056	0.398	0.097	−0.030	−0.018	0.098	0.31	7.26	
Notes.

*, ** Significant at P < 0.05, 0.01 levels, respectively (least significant difference test).

‘Total represents fitting RU and BRU and CK treatment simultaneously; ‘F’ represents F value at significance analysis.

Compared with a single soil layer (10-, 20-, or 40-cm), the soil N2O emissions can be predicted more accurately by combining the moisture contents and temperatures of the three soil layers [Eq. (6)]. The water contents and temperatures of the three soil layers affect the N2O emissions to different degrees. Therefore, we believe that the soil N2O emissions are due to the combined effect of the moisture content and temperature of the 0–40-cm soil layer. Moreover, the three treatments cannot be adequately fit with one regression equation (R2, 0.51 in 2019). When multivariate nonlinear fitting was applied to the three treatments, an R2 value above 0.60 was obtained (Fig. 4). The results show that the N2O emissions in the RU treatment are significantly higher than those in the BRU and CK treatments (Fig. 3), indicating that the soil environment (two or more variables) significantly affects the response of the soil N2O emissions to changes the moisture content and temperature. When the soil environment was changed by fertilization and biochar amendment, the accuracy of multivariate nonlinear fitting decreased significantly, as per dimensionality reduction analysis (MNF-DR). This is due to the changes in soil biochemical reaction rate due to fertilization or biochar amendment (Bruun et al., 2014; Saarnio, Heimonen & Kettunen, 2013) and changes in the response of the N2O emissions to moisture content and temperature.

Figure 4 The fitting about WFPS, soil temperature, and N2O emissions based on MNF-DR analysis for each treatment during 2019–2020 is presented.

(A) and (E) represents the measured data of the three treatments (RU + BRU + CK) we used for MNF-DR analysis in 2019 and 2020, respectively; (B) and (F) represents the measured data of RU we used for MNF-DR analysis in 2019 and 2020, respectively; (C) and (G) represents the measured data of BRU we used for MNF-DR analysis in 2019 and 2020, respectively; (D) and (H) represents the measured data of CK we used for MNF-DR analysis in 2019 and 2020, respectively.

Researchers have also suggested that the soil water content (or temperature) at a depth of five cm can be identified as the single trigger for N2O emission (Lognoul et al., 2019). We acknowledge that most N2O emissions originate from biochemical reactions in the topsoil, but some N2O emissions still arise from deep soil via nitrification and denitrification. The N2O produced in the deep soil will diffuse to the surface at a rate that depends on the soil moisture content and the N2O concentration gradient (Shcherbak & Robertson, 2019). Furthermore, storage fluxes occurred in the deep soil. Lognoul et al. (2019) assumed that storage fluxes were negligible. However, the soil moisture content and temperature in the 0–40 cm soil layer were significantly correlated with N2O emissions; therefore, physical factors from topsoil cannot be accurately identified as the single trigger for N2O emissions.

The predictions using MNF-DR are more stable than those based on the surface soil moisture content and temperature only. However, with MNF-DR analysis, only the N2O emissions at the farmland scale can be predicted from the perspective of statistics. The N2O emissions trends depending on changes microscale in the water content or temperature cannot be explained. Moreover, the relationship k4BRU >k4RU indicates that the N2O emission rate of BRU was greater than that of RU (Table 3). In fact, the N2O emissions rate of biochar treatment should be even lower. Thus, it is difficult to predict the N2O emissions trend of each treatments from this MNF-DR analysis. We need a more accurate model to predict the impact of biochar on the N2O emissions rate.

Establishing function among soil water content, and N2O emissions

The emission rate of N2O increases with increasing soil moisture content based on this field experiment, while the increase in the N2O emissions slows down after the WFPS exceeds 80%. Previous studies showed that the N2O emissions increase until the WFPS reaches ∼75% (Lan et al., 2013). However, the N2O emissions rate decreases when the soil water content exceeds a WFPS of 75% (Prado et al., 2006) because the anaerobic environment accelerates the reduction of N2O to N2 in the soil (Wu et al., 2013). The soil represents a N2O sink when the WFPS is below 25% (Flechard et al., 2007; Goldberg & Gebauer, 2008; Wu et al., 2013).

Thus, we assume that the rate of N2O emissions varies with the WFPS. The rate of N2O emissions was low at a low WFPS. and increased sharply with increasing WFPS. When the WFPS exceeds a particular value, the increase in the soil N2O emissions slowed down despite the continuous increase in the WFPS. The WFPS ranges from 0 to 1; the emitted N2O responds to a variation in the WFPS.

Figure 5 The fitting about WFPS and N2O emissions based on the exponential model for each treatment during 2019–2020 is presented.

(A) and (D) represents the measured data of RU we used for the exponential model in 2019 and 2020, respectively; (B) and (E) represents the measured data of BRU we used for the exponential model in 2019 and 2020, respectively; (C) and (F) represents the measured data of CK we used for the exponential model in 2019 and 2020, respectively.

The soil was amended with biochar at a depth above 20 cm and we fitted the model for the 20-cm soil layer to reduce the influence of soil water evaporation. For each treatment, the N2O emissions were plotted against the WFPS (Fig. 5). These values were fitted using Eq. (9), yielding a strong positive correlation (Table 4). Thus, Eq. (8) explains the N2O emissions well. The smaller the coefficient a is, the smaller the growth rate of N2O emission is (Eq. (9)). The relationship aBRU >aRU indicates that the N2O emissions rate of BRU is lower than that of RU (Table 4). Thus, after biochar application, N2O emissions increased significantly with increasing WFPS after fertilization (Fig. 6). To quantify the effect of biochar on the N2O emissions, the SC [Eq. (9)] was determined, which has been applied in many previous studies (Engel et al., 2017; Joby & Mahanthesh, 2019; Tan, Cui & Luo, 2017). The SC of RU versus CK (SCU) is 19.18 and 20.83 in 2019, respectively. and The SC of BRU versus CK (SCB) is 1.02 in 2019 and 14.74 in 2020, respectively. Thus, we can conclude that biochar significantly reduces the N2O emissions, which efficiently inhibits the N loss.

Table 4 N2O emission observations and MNF-DR analysis during the growing period of maize.

Coefficients, determinative factor, and F value of Eq. (9).

	Treatment	a	b	c	R2	F value	
	RU	−6.436*10−5	0.052	−6.218	0.70	153.90**	
2019	BRU	−1.266*10−4	0.053	−6.365	0.81	535.23**	
	CK	5.251*10−4	0.091	−6.898	0.85	411.30**	
	RU	−4.282*10−4	0.074	−4.774	0.78	587.31**	
2020	BRU	−7.811*10−4	0.131	−7.411	0.74	320.72**	
	CK	−6.791*10−4	0.110	−7.594	0.59	165.93**	
Notes.

*, ** Significant at P < 0.05, 0.01 levels, respectively (least significant difference test).

Figure 6 Determinants of soil N2O emissions.

Discussion

N2O is an intermediate product that is formed during both denitrification and nitrification (Dobbie & Smith, 2001). Soil moisture is the most critical factor governing N2O and NO formation when mineral N sources in soil are limited (Prado et al., 2006). N2O emissions have been reported to increase until the WFPS reaches ∼72% (Schmidt, Thöni & Kaupenjohann, 2000), while other have been some reported of a threshold reaching up to 90% (Dobbie & Smith, 2001). The positive correlation coefficient between the N2O emissions and WFPS obtained in our study implies that an increase in the soil water content promotes the soil N2O emissions (Table 1). The soil water content indirectly affects the soil N2O emissions because the volumetric gas content affected by the WFPS is a vital factor governing both nitrification and denitrification (Clough et al., 2017). Denitrification mainly occurs above a WFPS of 60–70%, whereas nitrification occurs at a WFPS of 35% and 60% (Bateman & Baggs, 2005). Most N2O originates from nitrification when the WFPS is below 60%, while an increased conversion from N2O to N2 occurred at higher soil water contents (Wu et al., 2013). Thus, the emission flux of the soil N2O decreases when denitrification was dominant, although the cumulative N2O emissions continue to increase (Figs. 5 and 6). This conclusion agrees with the results of other studies, in which a nonlinear N2O emission response to N fertilizer addition was reported (Clairep, 2005; Prado et al., 2006).

The soil N availability may have a significant impact on the N2O emissions. N fertilization, a direct measure of the increase in the soil N availability, promotes the N2O emissions compared with the unfertilized control (Lei, Ding & Cai, 2005).

Biochar application led to a significant increase in the soil water content in the topsoil (0–20 cm) relative to the unamended biochar treatments (Fig. 1). Many studies have been carried out to improve the soil water holding capacity, which should enhance the water use efficiency in agricultural production (Basso et al., 2013; Oki, 2006). Amendment with biochar significantly mitigates the soil N2O emissions, particularly at a WFPS above 60% (Fig. 6). Previous studies have shown that the N2O emissions from biochar-amended soil were sharply reduce because the biochar adsorbed inorganic N and thus decreased the N concentration available for nitrification and denitrification (Taghizadeh-Toosi et al., 2011; (Cayuela et al., 2014; Stewart et al., 2013). Moreover, the response of the decreased N2O emissions to temporary immobilization of available N was derived from a high C:N ratio after biochar amendment (Baggs, Watson & Rees, 2000). The decreased N availability due to biochar adsorption only partly explains the reduction of the N2O emissions compared with the BRU and RU treatments. Because of the strong correlation between the WFPS and soil N2O emissions, the amendment with biochar also mitigates the N2O emissions by increasing the soil water content (Table 1). This result is consistent with the finding that the anaerobic environment caused by a high soil water content increases the abundance of denitrifying bacteria and thus catalytically reduces N2O to N2 (Wu et al., 2013). Other studies have shown that biochar significantly increases the soil N2O emissions under increased N availability due to fertilization (Clough et al., 2010; Rajkovich et al., 2012). It is possible that incomplete nitrification occurs after amendment with biochar (Clough et al., 2017). Biochar increase surface soil temperatures, which in turn increases N2O emissions. The effect of temperature on N2O emission was due to the promotion of the microbial activity. High temperatures could also enhance the denitrifying bacteria activity, promoting the conversion of N2O to N2. Nevertheless, biochar only increases the surface soil temperature (0–10 cm). The model shows that N2O emission was affected by soil moisture (or temperature) above a depth of 40 cm depth. Therefore, a small temperature increase does not have a significant impact on N2O emissions.

Biochar enhances soil nitrogen and water immobilization, promotes crop photosynthesis, and increased crop yield (Macdonald et al., 2014; Zhao et al., 2014). Compared with straw returns, biochar can improve soil physical and chemical properties of soil, enhance the effectiveness of water and fertilizers, and reduce chemical fertilizer pollution. However, expensive straw carbonization equipment restricts the market development of biochar and its implication, and the production rate of carbonization equipment does not reach the standard of large-scale production (Zhang et al., 2019a). Therefore, low-cost and large-scale biochar production biochar is a challenge in developing agricultural ecology.

Conclusions

In this research, we have established an appropriate standard for evaluating soil C and N improvement benefits. MNF-DR analysis was difficult to predict the N2O emissions trend, especially when WFPS was above 70% (k4BRU > k4RU). The exponential model can accurately simulate the N2O emission trend based on WFPS (R2 > 0.59). That is, the emission rate of N2O initially increased and then decreased with the increased WFPS, which was consistent with the previous study. The relationship aRU < aCK indicates that fertilization did promote N2O emissions, while aBRU > aRU indicated that biochar applications mitigate the N2O emissions induced by fertilization (Table 4). Moreover, The SC of RU versus CK (SCU) was 19.18 and 20.83 in 2019 and 2020, respectively, while the SC of BRU versus CK (SCB) is 1.02 in 2019 and 14.74 in 2020, respectively, indicating that biochar significantly reduced the sensitivity of N2O to the substrate, which reduces the N2O emissions.

From the exponential model, we believe that the input of biochar and urea did not change the N2O emissions trend in dryland farming. Furthermore, the exponential model confirmed that biochar indeed reduced the N2O emissions flux induced by urea application. However, our study was based on the physical properties of biochar (water-holding capacity) to explore the greenhouse gas emissions pattern. When the added carbon source is organic matter, such as organic fertilizer and straw, the accuracy of this model needs to be further revised. In a word, we suggested that the exponential model can be used to quantify the impact of biochar and urea on soil N2O emissions in dryland farming.

Supplemental Information

Supplemental Information 1 Data

All birds which did not show normal growth in the first examination.

Click here for additional data file.

Additional Information and Declarations

Competing Interests

Author Contributions

Data Availability

The authors declare there are no competing interests.

Xiao Wang conceived and designed the experiments, performed the experiments, analyzed the data, prepared figures and/or tables, authored or reviewed drafts of the paper, and approved the final draft.

Ping Lu analyzed the data, prepared figures and/or tables, authored or reviewed drafts of the paper, and approved the final draft.

Peiling Yang performed the experiments, prepared figures and/or tables, authored or reviewed drafts of the paper, and approved the final draft.

Shumei Ren performed the experiments, authored or reviewed drafts of the paper, and approved the final draft.

The following information was supplied regarding data availability:

The raw measurements are available in the Supplemental File.

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
