# Peer review of "Effects of fertilizer and biochar applications on the relationship among soil moisture, temperature, and N2O emissions in farmland"

_PeerJ, doi:10.7717/peerj.11674_

## Round 0.1 · original submission · Major Revisions

'Effects of fertilizer and biochar applications on the relationship among soil moisture, temperature, and N2O emissions in farmland' needs improvements as in its present shape it is not appropriate. If authors can redesign the whole article by considering the following points it would be great.

1. Abstract needs to be quantitative
2. Introduction needs to be written by considering what is the problem and how it can be addressed by using different approaches and then approach in this study
3. Methodology needs to be clear with justification why and how
4. Results are not clear. Kindly consider one good article to see how to write results
5. Link results with previous work by giving reasons
6. Conclusion should give final suggestions

Reviewer 1 ·

Basic reporting

I have gone through the manuscript and i found it language unclear .Literature references, sufficient field background/context has not provided.

Experimental design

Research question is not well defined. Methods described with in sufficient detail .

Validity of the findings

Conclusions are not well stated, linked to original research question & limited to supporting results

Additional comments

I have gone through the manuscript entitled Effects of fertilizer and biochar applications on the relationship among soil moisture, temperature, and N2O emissions in farmland. The author has done work but the author has not represented his work in a perfect way. Already too much data are available on such issue and author should touch new dimension of this field.The language of the manuscript is difficult to understand.

·

Basic reporting

No comment

Experimental design

No Comment

Validity of the findings

No Comment

Additional comments

Comments in attached pdf

---

## Round 0.2 · Minor Revisions

The article is in better shape now and the authors have addressed reviewers' concerns. However, it is recommended that the author should delete excessive self-citation and add clear recommendations in the conclusion section for further evaluation of this work.

Reviewer 3 ·

Basic reporting

- Some references required in introduction
- Figures 4 and forward, caption requires more information to explain what the panels and symbols represent

Experimental design

N/A

Validity of the findings

N/A

Additional comments

- In line 50 (straw returning, biochar application, et al.), please check et al or etc?
- Line 62 lacks of additional references, you can use the following recent references for help
https://doi.org/10.1080/00380768.2020.1718923
https://doi.org/10.1016/S1002-0160(15)60073-X
https://doi.org/10.1038/s41598-021-88293-6
https://doi.org/10.1007/s42729-021-00409-z
- line 120 , please specify the soil depth
- Figures 4 and forward, caption requires more information to explain what the panels and symbols represent.

---

## Round 0.3 · accepted · Accept

The article can be accepted.